# *Escherichia coli* Aggregates Mediated by Native or Synthetic Adhesins Exhibit Both Core and Adhesin-Specific Transcriptional Responses

Yankel Chekli,[a] Rebecca J. Stevick,[a] Etienne Kornobis,[b,c] Valérie Briolat,[b,c] Jean-Marc Ghigo,[a] Christophe Beloin[a]

[a]Institut Pasteur, Université Paris Cité, CNRS UMR 6047, Genetics of Biofilms Laboratory, Paris, France
[b]Hub de Bioinformatique et Biostatistique-Département Biologie Computationnelle, Institut Pasteur, USR 3756 CNRS, Paris, France
[c]Plate-forme Technologique Biomics—Centre de Ressources et Recherches Technologiques, Institut Pasteur, Paris, France

**ABSTRACT** Bacteria can rapidly tune their physiology and metabolism to adapt to environmental fluctuations. In particular, they can adapt their lifestyle to the close proximity of other bacteria or the presence of different surfaces. However, whether these interactions trigger transcriptomic responses is poorly understood. We used a specific setup of *E. coli* strains expressing native or synthetic adhesins mediating bacterial aggregation to study the transcriptomic changes of aggregated compared to nonaggregated bacteria. Our results show that, following aggregation, bacteria exhibit a core response independent of the adhesin type, with differential expression of 56.9% of the coding genome, including genes involved in stress response and anaerobic lifestyle. Moreover, when aggregates were formed via a naturally expressed *E. coli* adhesin (antigen 43), the transcriptomic response of the bacteria was more exaggerated than that of aggregates formed via a synthetic adhesin. This suggests that the response to aggregation induced by native *E. coli* adhesins could have been finely tuned during bacterial evolution. Our study therefore provides insights into the effect of self-interaction in bacteria and allows a better understanding of why bacterial aggregates exhibit increased stress tolerance.

**IMPORTANCE** The formation of bacterial aggregates has an important role in both clinical and ecological contexts. Although these structures have been previously shown to be more resistant to stressful conditions, the genetic basis of this stress tolerance associated with the aggregate lifestyle is poorly understood. Surface sensing mediated by different adhesins can result in various changes in bacterial physiology. However, whether adhesin-adhesin interactions, as well as the type of adhesin mediating aggregation, affect bacterial cell physiology is unknown. By sequencing the transcriptomes of aggregated and nonaggregated cells expressing native or synthetic adhesins, we characterized the effects of aggregation and adhesin type on *E. coli* physiology.

**KEYWORDS** aggregates, *E. coli*, Ag43, nanobodies, bacterial physiology, anaerobia, stress tolerance, antigen 43, *Escherichia coli*, stress response, tolerance

Bacteria often live in communities called biofilms, adhering to each other and to biotic or abiotic surfaces. Autoaggregation, defined as bacterium-bacterium interactions of genetically identical strains (1), is observed in environmental and pathogenic species and contributes to biofilm formation (2, 3). The perception of the importance of aggregate formation has increased due to the awareness that, in most cases, natural and clinical biofilms resemble cell aggregates rather than large, highly structured biofilms (4, 5). Although rarely larger than a few hundred micrometers, these aggregates maintain biofilm characteristics, including enhanced tolerance of physical and chemical stresses and host immune defenses (6–8). In *Pseudomonas aeruginosa*, aggregation is induced upon exposure to detergent (9) and has been shown to promote

Address correspondence to Christophe Beloin, christophe.beloin@pasteur.fr.

The authors declare no conflict of interest.

*P. aeruginosa*'s survival of exposure to antimicrobial treatments in patients with cystic fibrosis (10, 11). Moreover, while *P. aeruginosa* and other species of bacteria, such as *Sphingobium* spp., use aggregation as a defense mechanism against predation by protozoa (8, 12), the formation of aggregates has also been shown to be an essential step that promotes *Neisseria meningitidis* virulence (13).

*Escherichia coli*, a versatile bacterium that can behave either as a pathogen or a commensal, is prone to aggregation mediated by a range of adhesins, including the autotransporter antigen 43 (Ag43). The self-recognition properties of Ag43 promote autoaggregation and biofilm formation (14–16). Ag43-dependent aggregates exhibit specific properties, including providing protection against neutrophil killing, promoting intracellular aggregate and biofilm formation, and contributing to the persistence and virulence of uropathogenic *E. coli* in the mouse bladder (15).

While it is well-accepted that aggregate formation correlates with stress tolerance, the underlying mechanisms allowing aggregated bacteria to grow or survive under adverse conditions remain unknown. Here, we used transcriptome sequencing (RNA-seq) and a specific setup of *E. coli* K-12 strains expressing either native (Ag43) or nanobody (Nb)-based synthetic self-recognizing adhesins to identify changes in gene expression associated with stress resistance between nonaggregated and aggregated cells. We found that, regardless of the adhesin mediating aggregation, the aggregates exhibited a core transcriptomic signature indicative of a specific metabolic rewiring and characterized by the activation of several stress resistance pathways. Beyond this core response to aggregation, we also showed that many genes were specifically regulated in Ag43-dependent aggregates compared to aggregates mediated by synthetic adhesins. These results demonstrate the existence of a specific physiological response to aggregation and suggest that aggregate physiology leads to profound metabolic changes, including the activation of multiple stress response systems.

## RESULTS

**Both native and synthetic adhesins promote equivalent *E. coli* autoaggregation and tolerance of antibiotic stress.** To better understand the impact of autoaggregation on *E. coli*'s physiology, we chose a combination of native and synthetic self-recognizing adhesins to investigate both general and adhesin-specific transcriptomic responses. Ag43 (also known as Flu) was used as the native adhesin since it is the major self-recognition and aggregation factor in *E. coli* K-12. Ag43 is a naturally phase-variable adhesin, being either in an on (Ag43+) or off (no Ag43) status. Ag43's on and off status was monitored through the use of an operon fusion between *ag43* and the *lacZ* gene (17). For this adhesin, we used a series of four strains (Fig. 1A). Two strains represented the aggregated state: (i) PcL*ag43*, constitutively expressing Ag43, and (ii) Ag43_On, containing cells in an Ag43 on state with native levels of Ag43. Two other strains represented the nonaggregated state: (i) a Δ*ag43* mutant with *ag43* deleted (no Ag43) and (ii) Ag43_Off, containing cells in an Ag43 off state.

For the synthetic adhesins, we used chimeric fusion proteins consisting of a translocator and an exposed domain (Fig. 1B) (18). The outer membrane anchor was composed of the intimin N terminus from an enterohemorrhagic *E. coli* (EHEC) strain, O157: H7, including a small N-terminal signal peptide, a LysM domain for peptidoglycan (PG) binding, and a $\beta$-barrel for transmembrane insertion (18, 19). The exposed domains, composed of a nanobody (the variable domains of camelid heavy-chain antibodies) or its corresponding antigen, were then fused to the C-terminal part of the truncated intimin. Recognition between a nanobody (Nb2 or Nb3) expressed on one *E. coli* cell and its corresponding antigen (Ag2 or Ag3) on another cell promotes aggregation. We swapped the *ag43* gene for the different synthetic-adhesin genetic constructs and placed them under the control of the constitutive lambda P_R promoter PcL. For the synthetic adhesins, the aggregated state was represented by a 1:1 mixture of cells expressing Nb2 and Ag2 or Nb3 and Ag3. The nonaggregated state was represented by a 1:1 mixture of cells expressing Nb2 and Ag3 or Nb3 and Ag2 (Fig. 1B).

We first verified the capacity of each adhesin to promote aggregation. After 6 h of incubation, strains PcL*ag43* and Ag43_On and cultures composed of Ag2/Nb2 and Ag3/Nb3

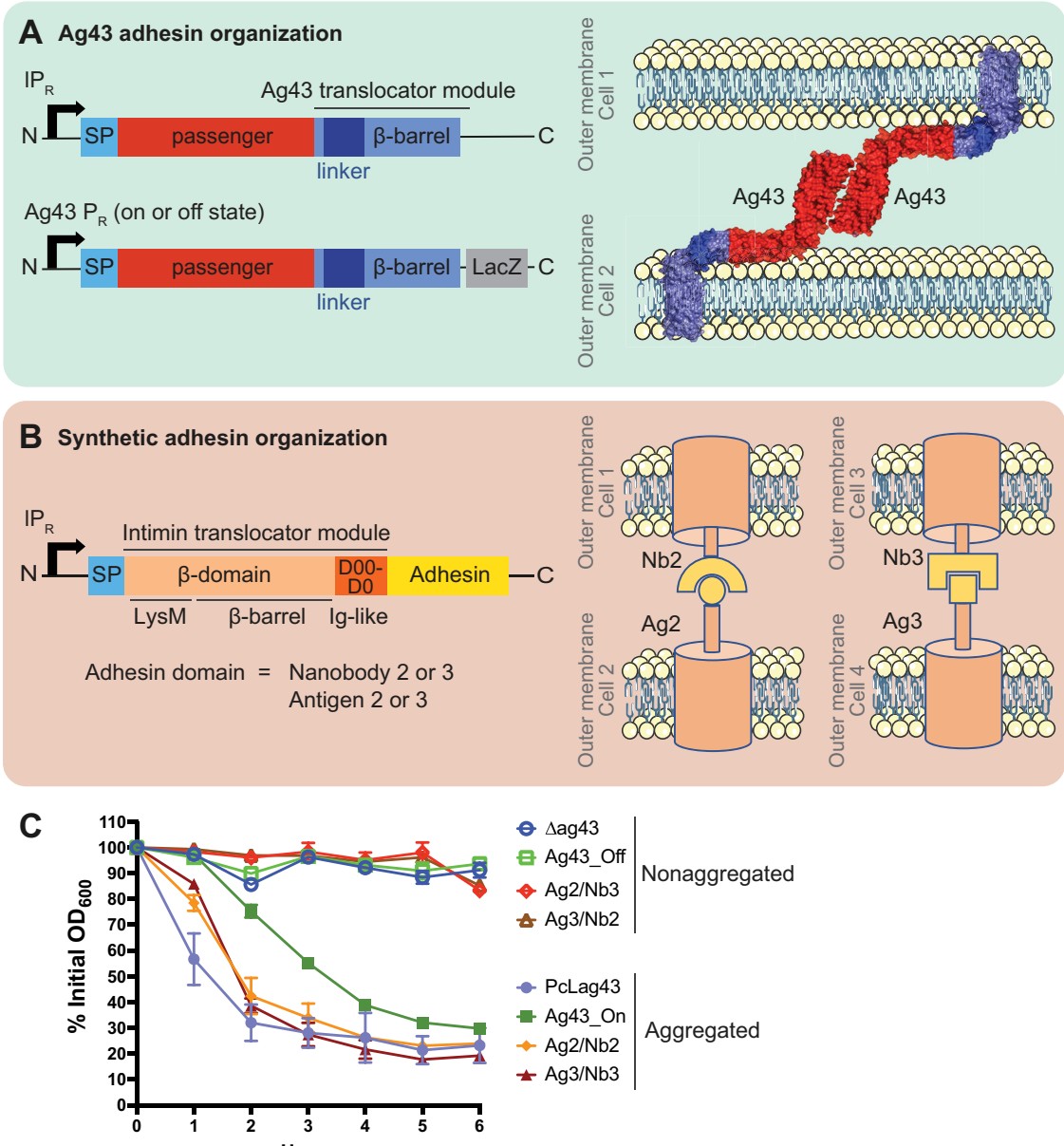

**FIG 1** Genetic organization and aggregation capacity of the adhesins used in this study. (A) Genetic organization of the native Ag43 adhesin constructs. (B) Genetic organization of the synthetic adhesin constructs. SP, signal peptide. In the different synthetic adhesin strains, the yellow part corresponds to nanobody 2 or its corresponding antigen 2 or to nanobody 3 or its corresponding antigen 3. (C) Aggregation curves performed with all the strains and strain couples used in this study. Results are expressed as the percentage of the $OD_{600}$ measured at the top of the tube at time zero. For each strain, measurements were taken at the exact same position in the tube throughout the time of the experiment. A higher percent OD measurement corresponds to no aggregation, and a decrease in percent OD indicates aggregate formation. Plotted data represent the mean values ± standard deviations from 3 biological replicates, and each replicate is the mean of 2 technical replicates.

provided similar levels of aggregation, while the Δ*ag43* (Ag43_Off strain) and cultures composed of Ag2/Nb3 and Ag3/Nb2 displayed no aggregation (Fig. 1C). Measurements of the aggregate sizes in culture also revealed that, although the sizes of Ag2/Nb2, Ag3/Nb3, and Ag43_On aggregates were comparable, aggregates made with PcL*ag43* strains were almost 10 times bigger, suggesting that overexpressing Ag43 led to stronger interactions, resulting in an increased aggregate size (Fig. S1 in the supplemental material). We then assessed the capacity of the different aggregated cells to sustain a lethal antibiotic stress compared to that of their nonaggregating counterparts. Aggregated cells survived the lethal action of amikacin, an antibiotic recommended for the treatment of *Enterobacterales*

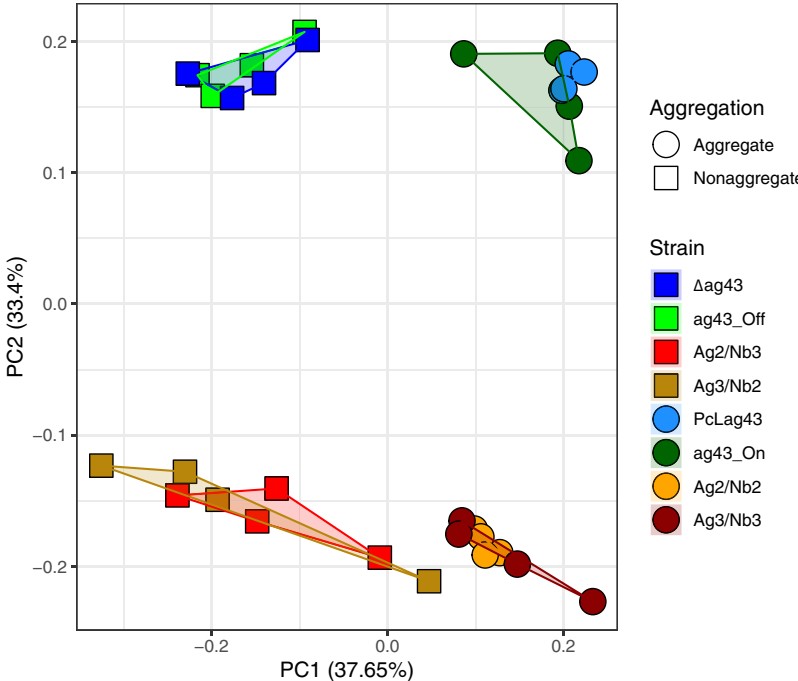

**FIG 2** Aggregation capacity and adhesin type affect overall transcriptional profiles. Principal-component analysis (PCA) of all samples sequenced (4 biological replicates for each of the 8 conditions) based on variance-stabilizing transformation (VST)-normalized transcript counts. Colors represent the different strains, while shapes represent the aggregation status.

infections (20–22), much better than nonaggregated cells, therefore validating our setup to evaluate the corresponding aggregate physiology (Fig. S2).

**Autoaggregation leads to a robust core transcriptional response independently of the adhesin type.** In order to perform comparative RNA-seq, we developed a protocol to optimize the separation between planktonic cells and aggregates. For this, different strains and synthetic mixtures were independently cultured in Miller's lysogeny broth (LB) to an optical density (OD) of 0.5 and then left to aggregate in separating funnels for 3 h at 37°C under static conditions (Fig. S3). One milliliter of cells was sampled from the bottom of each funnel to perform RNA extraction and a comparative transcriptomic analysis of the different types of cultures.

The overall transcriptional profiles of each condition clustered depending on the strains' capacity to aggregate and the type of adhesin expressed (natural Ag43 or synthetic nanobodies), as shown in the principal-component analysis (PCA) calculated using normalized transcript counts (Fig. 2). The first principal component (PC1), explaining 38% of the variance, separated the samples based on aggregation, regardless of the type of adhesin. Native and synthetic aggregates clustered along the *x* axis, with nonaggregated controls also clustered based on PC1. On the other hand, the second principal component (PC2), explaining 33% of the variance, separated the samples based on their adhesin type, with the samples expressing native adhesins being very close on the *y* axis and separate from the bacteria expressing synthetic adhesins. Thus, while we can see that the aggregates, native and synthetic, are close to each other according to PC1, suggesting a core response of the bacteria toward aggregation, we can also observe that the samples are different according to PC2. This illustrates that, in addition to aggregation, the type of adhesin used, native or synthetic, is a discriminating factor, suggesting a specific response of bacteria depending on the adhesin.

The *E. coli* K-12 genome has 4,401 genes encoding 116 RNAs and 4,285 proteins (23). In order to compare between nonaggregates and aggregates, differential expression analysis was conducted for each nonaggregating strain in comparison to the 2 aggregating strains of the same adhesin type (native or synthetic) (Fig. 3, bar plot on the left side). Considering these 8 different comparisons independently, we found that the pan response during

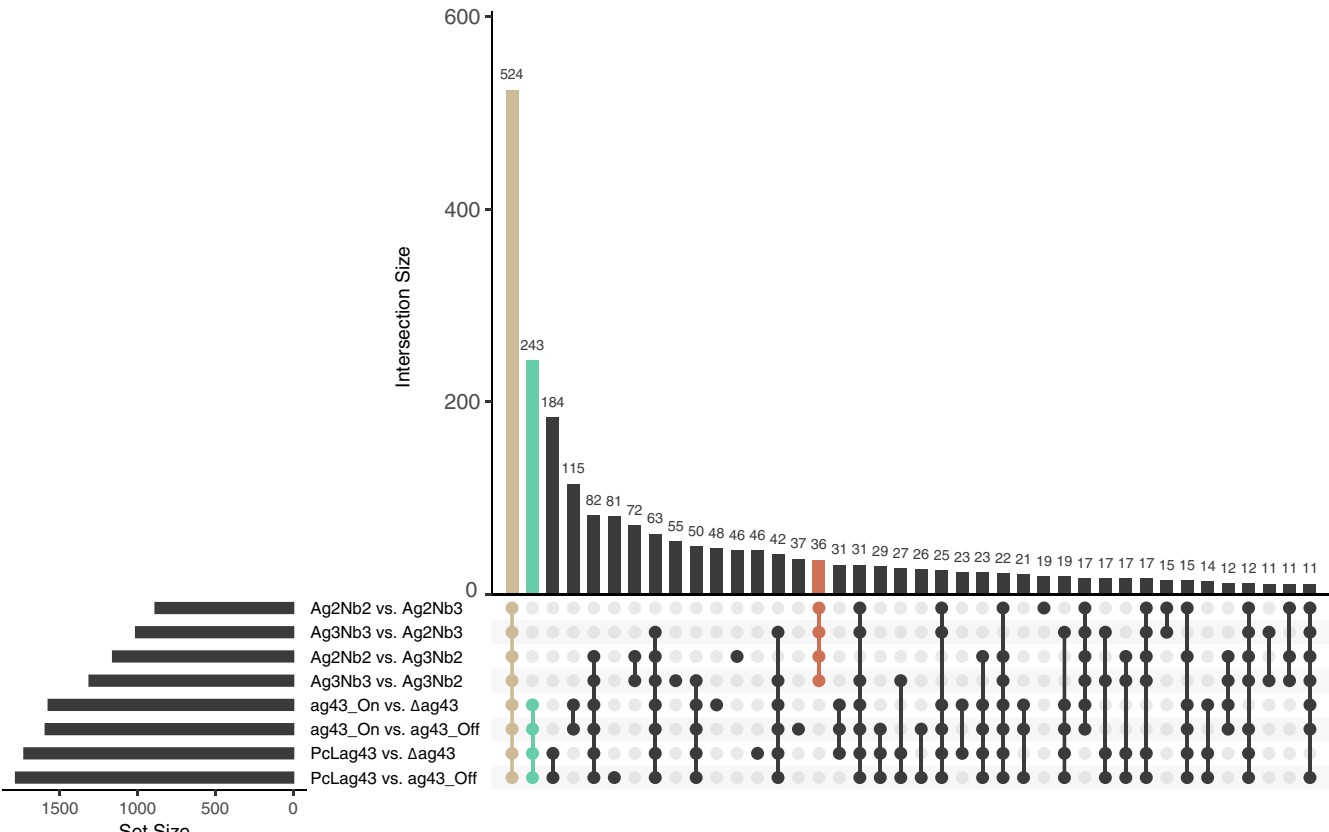

**FIG 3** The core transcriptomic response of bacteria to aggregation is independent of the type of adhesin mediating aggregation. Upset plot (Venn diagram alternative) showing the number of significantly differentially expressed genes shared between each comparison (vertical bars). The total number of genes significantly differentially expressed in each comparison is shown in the horizonal bar graph on the left. The genes differentially regulated as the core aggregate versus nonaggregate response are in beige. The genes differentially regulated specifically in Ag43-mediated aggregates are in green. The genes differentially regulated specifically in nanobody-mediated aggregates are in red. Genes were considered significantly differentially expressed when the Log$_2$ fold change was ≥1 or ≤−1 and the adjusted *P* value was <0.05.

aggregation was composed of 2,504 genes (56.9% of all coding genes) that were significantly differentially expressed upon aggregation for any adhesin type (considered significant when there was both a Log$_2$ fold change of ≥1 or ≤−1 and an adjusted *P* value of <0.05) (Table S1). This indicates that aggregation leads to a profound transcriptional reprogramming of the bacteria. Of these 2,504 genes, 1,380 were downregulated (31.3% of all coding genes) and 1,147 were upregulated (26.0%) (23 genes were either up- or downregulated depending on the comparison).

From these 8 different comparisons, we then identified the core response of *E. coli* toward aggregation, i.e., the genes that were up- or downregulated in all of the 8 comparisons of aggregates versus nonaggregates. This aggregation core response contained 524 genes (11.9% of the coding genes, with 208 upregulated [4.7%] and 316 downregulated [7.2%]) (Fig. 3, beige bar; Table S2) and corresponded to the group sharing the highest number of significantly regulated genes. This emphasized that there was a common response of bacterial aggregates formed upon the expression of native and synthetic aggregates.

**Aggregation leads to profound changes in metabolism and a global downshift of essential cellular functions.** To characterize the global effects of aggregation on biological processes, we used the PANTHER bioinformatics web server (http://www.pantherdb.org) (24) to determine the Gene Ontology (GO) groupings that were over- or underrepresented among significantly regulated genes in the core aggregate versus nonaggregate response. According to the number of significant genes, the overrepresentation assay determined whether the pathway was significantly different than would be expected by chance. The result is expressed as fold enrichment. Multiple pathways were found to be

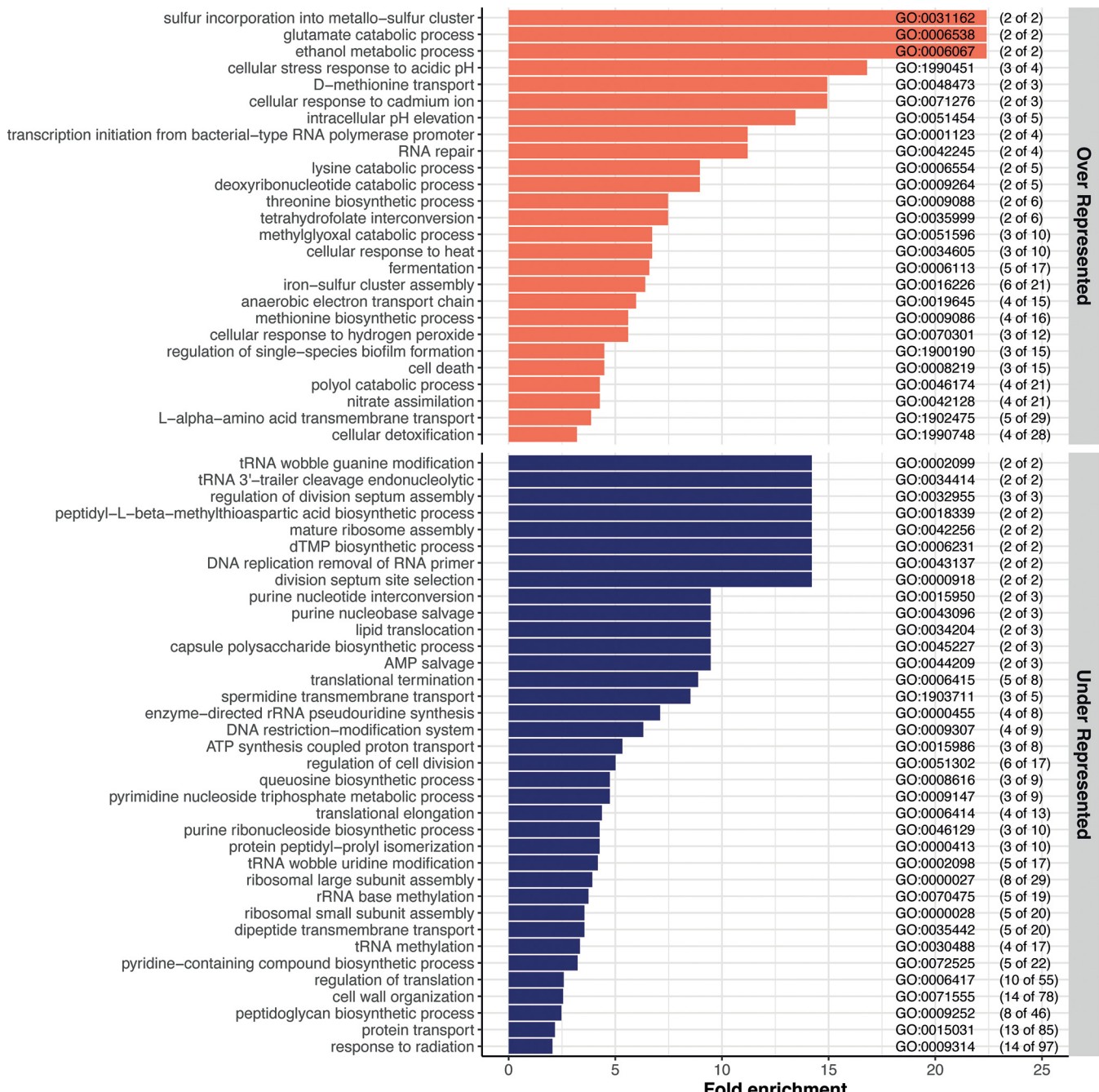

**FIG 4** Visualization of the GO biological processes significantly over- and underrepresented in all aggregate versus nonaggregate comparisons. Top, overrepresented pathways (orange bars); bottom, underrepresented pathways (blue bars). The GO enrichment analysis was performed via PANTHER using the biological process (BP) category. The genes included in this analysis are the 524 genes commonly regulated in all the comparisons of aggregates versus nonaggregates (Fig. 3, beige bar). The GO term for each biological process pathway is indicated on the right. The numbers in parentheses indicate the number of genes significantly up- or downregulated in the aggregates out of the total number of genes in the pathway.

significantly overrepresented (26 pathways) or underrepresented (36 pathways) in aggregates compared to nonaggregates (Fig. 4; Table S3 and Text S1).

(i) **Anaerobic metabolism.** Several of the overrepresented pathways were linked to a switch from aerobic to anaerobic metabolism (Table S3, pink-highlighted pathways). For example, we found that, upon aggregation, pathways related to fermentation, anaerobic respiration chains, and nitrite reduction were significantly enriched. Some genes classically induced under anaerobia (or microaerophilic conditions) and indicative of amino acid metabolism modification were also induced upon aggregation,

such as *adhE*, which catalyzes the reduction of acetyl-coenzyme A (CoA) to ethanol, and *cadAB* and *lcdC*, which are involved in lysine decarboxylase pathways.

**(ii) Amino acid metabolism, nucleotide production, and cell growth.** We detected additional modifications of amino acid metabolism, with the most overrepresented pathways corresponding to methionine biosynthesis/transport, threonine biosynthetic process, and glutamate catabolic process (Table S3, yellow-highlighted pathways). This reduction of essential growth functions was further supported by the downregulation of functions linked to ribosome assembly and functioning (Table S3, green-highlighted pathways), DNA replication and division (Table S3, orange-highlighted pathways), and pathways involved in the synthesis and/or salvation of purines, as well as pyrimidine synthesis. This decrease in nucleotide synthesis genes and recycling of nucleotides, as well as the downregulation of both RNase HI and HII, could lead to a decrease in the fidelity and the rate of DNA replication and, therefore, a decrease in the growth rate of bacteria. Additionally, three pathways involved in the regulation of cell division and positioning and assembly of the division septum were underrepresented. Finally, small and large ribosomal subunit assembly pathways were underrepresented in aggregates, along with ribosome assembly factors and RNA helicases. This reduction of ribosomal function might also be linked to induction of the stringent response with multiple (p) ppGpp alarmone-activated genes.

**(iii) Modification of cell envelope.** Cell envelope synthesis pathways, especially peptidoglycan (PG) synthesis pathways, were greatly impacted by aggregation (Table S3, violet pathways). This is consistent with an underrepresentation of pathways involved in membrane expansion that also indicates decreased metabolism, growth, and division. For instance, several pathways related to the cell wall were underrepresented: lipid translocation, capsule polysaccharide biosynthetic process, cell wall organization, peptidoglycan biosynthetic process, and protein transport. This supports the important modification of PG biogenesis and turnover that might be present in aggregated cells. In addition to PG-related genes, four of the seven genes involved in the lipoprotein posttranslational modification pathway were downregulated (*lgt*, *ispA*, *lolC*, and *lolD*) (Table S2), also suggesting a modification of the lipoprotein composition of the membranes upon aggregation.

**(iv) Biofilm-related genes.** Since aggregates can be considered early-stage biofilms, we also focused on genes known to be involved in biofilm phenotypes. Some of these were among the most upregulated genes in aggregates: *bhsA*, *bssS*, *bssR*, and *bolA*. *bhsA* has been shown to be induced by multiple stresses, and it seems to promote or reduce biofilm formation depending on the growth medium (25, 26). The expression of *bssS* and *bssR* was also shown to be increased in biofilms (27), and their associated proteins repress the motility of cells in LB, the medium used in our study (28). Therefore, increased expression of these genes may lead to reduced motility upon aggregate formation to avoid dispersal. Consistently, *bolA*, a motile/adhesive transcriptional switch involved in the transition from the planktonic state to a biofilm state, is also induced in aggregates (29).

Taken together, these results indicate that aggregation leads to a profound metabolic rewiring together with the reduction of several essential functions, indicating a global reduction of growth.

**Aggregates respond transcriptionally to different stresses.** The reduction of growth upon aggregate formation could be caused by stressful conditions developing within the aggregates. Accordingly, many genes related to various stress responses were induced upon aggregation (Table S3, pathways highlighted in red). Two of four acid resistance systems, GadABC and CadAB/LcdC, along with other acid stress response genes (*yhiM*, *hdeAB*, *hchA*, and *dps*), were enriched in aggregates. We also identified several genes associated with oxidative stress, such as hydrogen peroxide (*ychH*, *ygiW*, and *katG*), and with cellular detoxification (*katG*, *sodB*, *hcp*, and *frmA*) or protection (*dps*) that were significantly overexpressed in aggregates. This strengthens the hypothesis that certain bacteria within the aggregates maintain an aerobic metabolism. In addition to acidic or oxidative stress responses, we found several upregulated genes involved in the response

to heat (*clpA*, *clpB*, and *rpoH*) and general stress response (including *rpoS* and *rpoS*-activated genes). Finally, multiple genes that are induced or repressed by (p)ppGpp were identified as significantly regulated upon aggregation, suggesting an increased (p)ppGpp level from starvation and activation of the stringent response. These results indicate that cells forming aggregates were subjected to various types of stresses and activated multiple pathways to counteract them.

**Aggregation response is associated with antibiotic tolerance.** Consistent with previous studies, we determined that aggregated cells displayed enhanced survival when exposed to a lethal antibiotic stress (Fig. S2). While this increased tolerance could be linked to reduced growth and the different stress responses induced, we detected at least three other specific systems that were impacted in aggregates compared to nonaggregated cells and could be involved in sustaining antibiotic stress: tRNA modification and processing, toxin-antitoxin systems, and iron-sulfur systems.

**(i) tRNA modification and processing.** Four different pathways linked with the modification and processing of tRNAs were underrepresented in aggregates: tRNA wobble guanine modification, tRNA wobble uridine modification, tRNA 3′-trailer cleavage, and tRNA methylation (Table S3, blue-highlighted pathways). Additionally, other tRNA modification genes, as well as numerous genes encoding proteins that modify 16S RNA, 23S RNA, or ribosomal proteins, were also downregulated in aggregates (Table S2). Decreased expression of these different translational modifiers might increase the rate of mistranslation and, thus, enhance tolerance toward several antibiotics (30). Notably, some recent works have shown that specific tRNA modifications could be directly involved in enhanced tolerance of aminoglycosides (31, 32).

**(ii) Toxin-antitoxin systems.** We found that pathways involved in cell death and the production of toxic compounds were enriched upon aggregation, as well as finding upregulation of genes encoding toxins belonging to toxin-antitoxin (TA) systems. Since some of the toxins of these TA systems can act on different essential cellular processes, such as cell wall synthesis, membrane integrity, replication, transcription, translation, or cytoskeleton formation, they might modify the tolerance of bacteria toward antibiotics, although formal demonstrations of such effects are still missing (33–43).

**(iii) Iron-sulfur cluster systems.** We observed that the Suf system was induced upon aggregation (*sufABCDSE*). *E. coli* uses two systems to assemble iron-sulfur clusters: the Ics and the Suf systems (44). The Ics system is used under normal growth conditions, while the Suf system is employed under conditions of iron depletion or oxidative stress. The *suf* genes are induced by superoxide generators and hydrogen peroxide (45), further indicating that aggregated cells are subjected to multiple stresses. Interestingly, cells using the Suf system display enhanced tolerance toward aminoglycosides through a reduced proton motive force (PMF) that decreases entry of the antibiotics into the cells (46). This mechanism might be at play to explain the enhanced amikacin tolerance observed in our aggregates (Fig. S2).

Altogether, these results demonstrate the existence of a robust and mainly anoxia- and stress-related transcriptional response upon aggregation independent of the type of adhesin.

**Aggregation mediated by native *E. coli* adhesin leads to a specific transcriptional response with enhanced anaerobia and cell growth signatures.** While we identified a robust core transcriptomic response upon aggregation, we also detected responses specific to the adhesins used to mediate aggregation (Fig. 2) by comparing all the genes that were differentially regulated only in the native Ag43-mediated aggregates or only in the synthetic nanobody-mediated aggregates (Fig. 3, green and red bars). Clustering the genes using the GO database and enrichment analysis allowed us to identify differences in regulatory pathways between the two modes of aggregation.

In aggregates mediated by *E. coli* native Ag43 adhesin, we found that 243 genes (5.6% of coding genes) were uniquely up- or downregulated compared to their expression in nonaggregated controls (Fig. 3, red bar). Of these 243 genes, 122 were upregulated (2.8%) and 121 were downregulated (2.8%) (Fig. 5, representing the pathways associated with these genes; Tables S4 and S5). For the nanobody-mediated aggregates, the

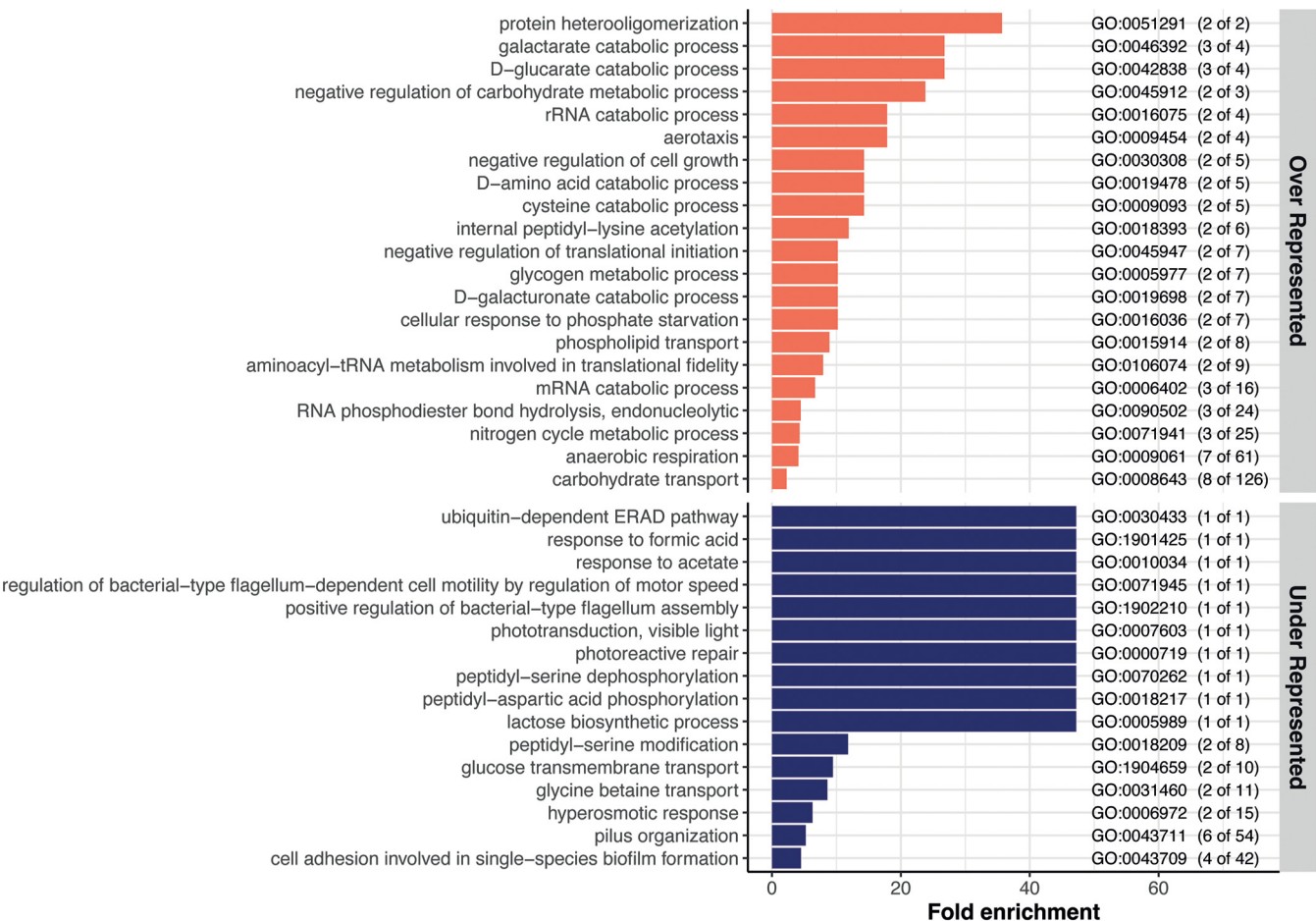

**FIG 5** Visualization of the significantly over- and underrepresented GO biological processes differentially regulated only in Ag43-mediated aggregates. Top, overrepresented (orange bars) pathways; bottom, underrepresented pathways (blue bars). The GO enrichment analysis was performed via PANTHER using the biological process category. The genes included in this analysis are the 243 genes commonly regulated in all the Ag43-mediated aggregates (Fig. 3, red bar). The GO term for each biological process pathway is indicated at the right. The numbers in parentheses indicate the numbers of genes significantly up- or downregulated in the aggregates out of the total number of genes in the pathway. Note that GO:0030433 for the ubiquitin-dependent ERAD pathway contains gene *ybeT*, which is a prokaryotic homolog of a eukaryotic Sel1 repeat-containing protein.

number of specific and differentially regulated genes was much smaller, with only 36 genes (0.8%) that were up- or downregulated compared to their expression in the nonaggregated control (Fig. 3, green bar). Of these 36 genes, 17 were upregulated (0.38%) and 19 were downregulated (0.43%) (Fig. 6, representing the pathways associated with these genes; Tables S6 and S7).

Beyond the specific induction of genes involved in catabolic processes of some sugar acids (*garK*, *garL*, *garR*, *uxaA*, and *uxaC* for D-glucarate, galactarate, and D-galacturonate) or D- and L-cysteine catabolism (*dcyD* and *yhaM*), genes encoding enzymes only active during anaerobiosis were also upregulated. These genes include *yhaM*, which codes for the major anaerobic cysteine-catabolizing enzyme (47), *ygfT*, involved in the catabolic process of urate (48), and genes involved in different aspects of anaerobic respiration (*frdD*, *napC*, *nirD*, *glpA*, *torA*, *ynfE*, and *dmsA*). This could indicate that Ag43-mediated aggregates display an enhanced level of anoxic conditions compared to nanobody-mediated aggregates. Additional toxin-antitoxin genes, with notably, the *mazEF* TA module, *relE* and *chpB*, were also differentially expressed only in Ag43-mediated aggregates. Interestingly, the three toxins MazF, RelE, and ChpB impair translation by degrading RNAs, suggesting that Ag43-mediated aggregates also display an enhanced reduction of translation. While only a few tRNA-encoding genes were downregulated in the core response to aggregation (7 genes), an important number of other tRNAs were specifically downregulated in Ag43-mediated aggregates (24 genes) (Tables S2 and S4).

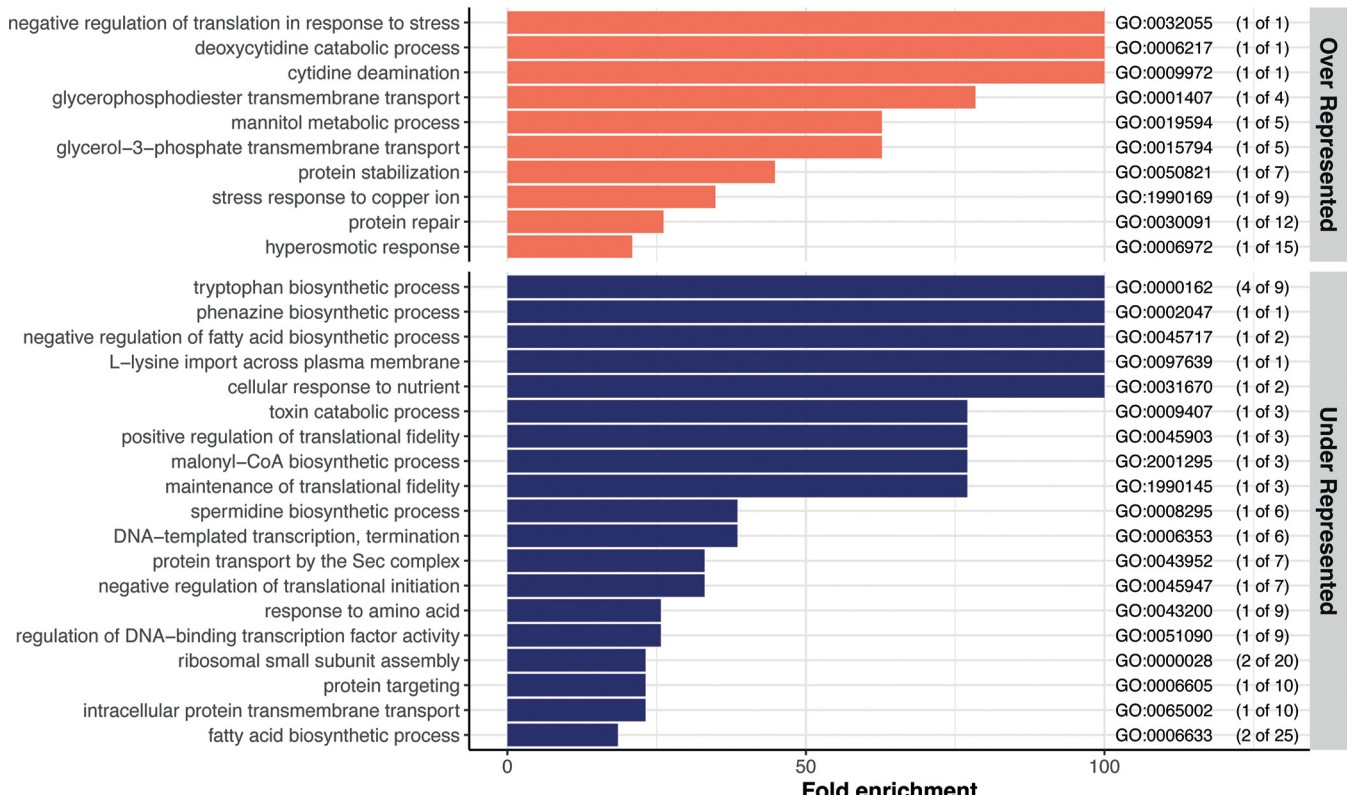

**FIG 6** Visualization of the significantly over- versus underrepresented GO biological processes differentially regulated only in nanobody-mediated aggregates. Top, overrepresented (orange bars) pathways; bottom, underrepresented pathways (blue bars). The GO enrichment was performed via PANTHER using the biological process category. The genes included in this analysis are the 36 genes commonly regulated in all the nanobody-mediated aggregates (Fig. 3, green bar). The GO term for each biological process pathway is indicated at the right. The numbers in parentheses indicate the numbers of genes significantly up- or downregulated in the aggregates out of the total number of genes in the pathway.

Finally, native aggregates are characterized by underrepresented genes related to motility and surface appendages. Multiple genes related to pili/fimbriae, known to counteract aggregation (49, 50), were downregulated (*htrE*, *sfmH*, *yadC*, *yadK*, *yehA*, *yehC*, and *yehB*). The impact on motility was less clear, since both *bdm*, a gene known to promote motility (51), and *ycgR*, encoding a phosphodiesterase described to impair motility (52), were simultaneously downregulated in Ag43-mediated aggregates. In contrast, synthetic-nanobody-mediated aggregation generated a limited specific response in addition to the core response. We detected the upregulation of genes encoding the small protein chaperone IbpB and the ribosome modulation factor Rmf, together with downregulation of genes encoding some ribosomal proteins (*rpsD*, *rpsK*, and *rpsM*) that might indicate some elevated level of stress and transition to a lower growth rate. We also measured a clear reduction of the expression of genes encoding enzymes responsible for the synthesis of tryptophan (*trpA*, *trpB*, *trpC*, and *trpD*).

## DISCUSSION

The formation of bacterial aggregates has significant relevance in different types of natural environments, as well as in clinical settings. To better understand the physiology of bacterial aggregates, we determined the transcriptomic response of *E. coli* to aggregation and the specificity of this response regarding the adhesin that mediates aggregation. We have shown that aggregation led to a significant and adhesin-independent core transcriptome and, therefore, a physiologic response in aggregated bacteria compared to nonaggregated bacteria. The transcriptional profile of the aggregated bacteria indicates a response to anoxia and other stressful conditions created by aggregation. In addition, we observed that many pathways involved in protein synthesis, DNA replication, cell division,

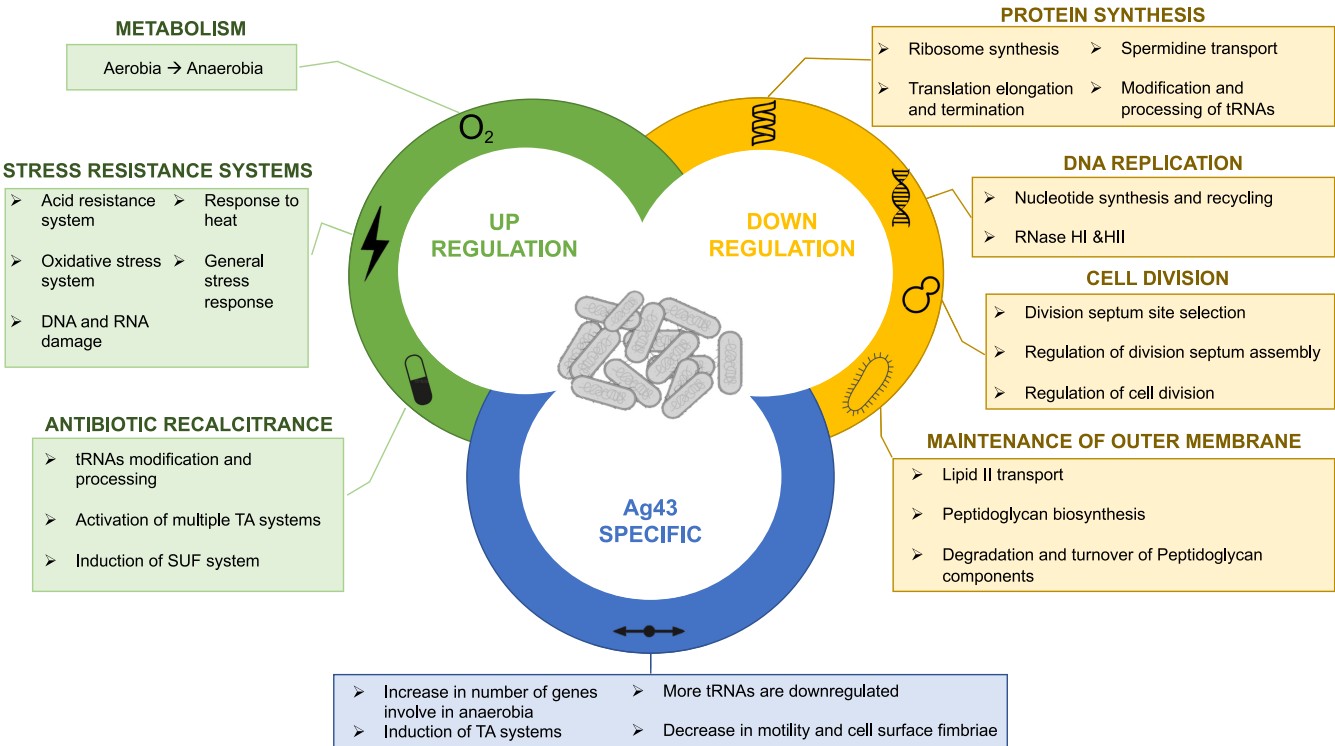

**FIG 7** Summary of the different biological processes regulated during *E. coli* autoaggregation. On the top are represented the main up- and downregulated biological processes, in green and yellow respectively, belonging to the core response of *E. coli* toward autoaggregation (524 genes total). At the bottom are represented the main changes specifically found in aggregates formed via the native *E. coli* adhesin Ag43 (243 genes). The genes specifically regulated within aggregates formed by synthetic adhesins are not shown in this figure.

and outer membrane maintenance were strongly downregulated in aggregates, indicating reduced bacterial growth. Furthermore, we found that many stress resistance systems, including antibiotic tolerance and acid stress, were upregulated, correlating with the fact that aggregates can withstand stressful conditions. Finally, we also observed that many genes were differentially regulated only when the aggregates were mediated by a native *E. coli* adhesin (Ag43), essentially extending the aggregation core response. This is indicative of a more complex and potentially time-adapted response when aggregates are formed via proteins that have been naturally present and coevolved with their host bacterial genomes. These results are summarized in Fig. 7.

Aggregation is often considered an early phase of biofilm formation, and aggregates themselves are fully integrated in the definition of biofilms. It has been shown previously that spontaneous 48-h-old aggregates of *P. aeruginosa* displayed properties resembling those of biofilms, such as reduced growth rate, enhanced tolerance to antimicrobials, or reduction of phagocytosis (53). Similarly, 24-h-old *P. aeruginosa* aggregates on alginate beads presented a transcriptional response indicating anoxic conditions (54). Anoxia is also a well-known condition within biofilms formed by facultative aero-anaerobic bacteria, such as *P. aeruginosa* and *E. coli* (55). This is due to the fact that the bacteria at the periphery of the biofilm actively consume oxygen, gradually reducing the oxygen levels at the center of the biofilm (55). Our results indicate that anoxic conditions are also present in young (3-h) *E. coli* aggregates. Hence, although aggregates are considerably smaller structures than biofilms, this rapid induction of anaerobia observed in our study may be due to similar mechanisms.

We and others have previously shown that one of the major physiological responses of bacteria within biofilms is the induction of stress responses (26, 56, 57). The induction of stress responses is a consequence of the physicochemical conditions prevailing within the biofilm environment. We show in this study that aggregation results in the induction of multiple stress resistance systems (e.g., acid, oxidative, heat shock, copper ion, and

cadmium), toxin-antitoxin systems, and starvation-related genes, along with a decrease in cellular functions related to bacterial growth. Interestingly, 24-h-old *P. aeruginosa* aggregates on alginate beads also displayed some signs of stress and reduced growth, with induction of heat shock genes and repression of genes related to ribosome biogenesis and functioning (54). The tolerance to various stresses observed in aggregates would therefore not be directly related to the experimental conditions, timing, or species *per se* but would rather be due to the physiology of aggregated bacteria. Although aggregates are smaller than biofilms, contain fewer bacteria, and do not produce a large amount of extracellular matrix, they still possess many biofilm properties, including their increased tolerance of antimicrobial agents. Interestingly, slow growth, starvation, and stresses that are encountered by aggregated cells have for long been described as triggering aggregation (58). This suggests that in a natural environment, these triggers might be maintained during and after the formation of aggregates, therefore allowing the cells to sustain and amplify a specific physiology to resist the surrounding stresses.

Many studies show that bacteria are able to sense contact with abiotic surfaces and that attachment to surface structures or deformation of the cell envelope results in physiological responses (59–63). These responses are mainly mediated by the activation of two-component systems and the production of secondary messengers, such as c-di-GMP. While we did not detect a clear modification of the expression of two-component-system sensors or regulators, we cannot exclude the possibility that some phosphorelay systems were induced or shut down upon aggregation. Several diguanylate cyclase- or phosphodiesterase-encoding genes were downregulated in aggregates, and while it is difficult to infer the exact net results of these changes in regulation on c-di-GMP levels, it is possible that signaling pathways could be altered during aggregation.

Whether the transcriptional response to aggregation is due to bacterial-bacterial interaction and/or surface sensing or whether it is the aggregate environment that influences the transcriptome of bacteria or both is therefore still an open question. It is indeed possible that the contact between bacteria leads to an early response that favors aggregation by decreasing motility and the expression of other surface structures. This switch is also characteristic of early biofilm formation, notably through c-di-GMP signaling (64). Subsequently, the environment within the aggregates would induce a secondary response, including the induction of anaerobic metabolism and the expression of multiple stress resistance systems. Further studies will be necessary to test these hypotheses.

If bacteria-bacteria interactions signal the transcriptional shift, one can then wonder whether the type of adhesin involved in the contact influences this signaling. The enhanced transcriptional response observed when using the native adhesin Ag43 instead of the synthetic ones suggests that this is the case, beyond a core response to aggregation. This Ag43-specific response did not correspond to new functions or pathways but seemed to reinforce the pathways already up- or downregulated in the core response to aggregation. Whether this enhanced response is a direct consequence of specific interactions mediated by Ag43 and/or a response to a different global architecture of the aggregates is unknown. Nevertheless, this suggests that the evolution that led to the generation of self-recognizing proteins was also associated with an adaptation of aggregation-mediated signal transduction. It would therefore be interesting to evaluate the transcriptomic response of aggregates mediated by native adhesins other than Ag43, to see if we observe a similar type of response.

Like Ag43, other types of adhesins, such as the Va-autotransported adhesins like TibA or AIDA-I (self-associating autotransporters [SAATs]) (65), trimeric autotransporters (TAAs), conjugation pili (66), or fimbriae, including curli or type IV pilus, can mediate aggregate formation through homotypic interactions and thus participate in kin recognition (67–75). Although all these cell surface appendages participate in kin recognition, they might trigger different transcriptional responses upon aggregation. Alternatively, exploring transcriptomic changes upon coaggregation, which refers to interactions between genetically distinct bacteria (76, 77), that can be mediated via heterotypic interactions between SAATs or TAAs in

closely related *E. coli* strains (16, 65, 78) may show different changes than those seen in autoaggregation.

In conclusion, this study provides new insights into the specific properties and physiology of aggregates and opens the way to the characterization of adhesin-specific responses that could be extended to other systems for a better understanding of the functions allowing enhanced tolerance of bacteria to environmental stress.

## MATERIALS AND METHODS

**Bacterial strains and growth conditions.** The bacterial strains used in this study are listed in Table S8. All strains were grown in Miller's lysogeny broth (LB) (Corning), supplemented with chloramphenicol (25 $\mu$g/mL), kanamycin (50 $\mu$g/mL), or ampicillin (100 $\mu$g/mL) when needed. Cultures were incubated at 37°C with 180-rpm shaking during growth phase or at 37°C under static conditions during the aggregation phase. Cultures on solid medium were done on LB with 1.5% agar supplemented with antibiotics when required. Prior to liquid cultures, bacteria were always streaked on LB agar from glycerol stocks. All media and chemicals were purchased from Sigma-Aldrich unless mentioned otherwise. All experiments and genetic construction were done in *E. coli* strain K-12 substrain MG1655 (F$^-$ $\lambda^-$ *rph-1*) obtained from the *Coli* Genetic Stock Center (CGSC#6300).

**Strain construction.** Insertions of the cassette carrying the chloramphenicol resistance gene and the $\lambda$P$_R$ or the synthetic adhesin construct were done as follows. First, the *E. coli* K-12 MG1655 strain was transformed with the pKOBEGA plasmid, which contains the $\lambda$-Red operon under the control of the arabinose-inducible *araBAD* promoter. After inducing the expression of the $\lambda$-Red genes, electrocompetent cells were prepared from this strain. In parallel, the CmPcL cassette and the synthetic adhesin constructs were amplified by PCR (PCR master mix, product number F548; Thermo Scientific) using long floating primers that carried 40 bp of homology with the insertion region at each end. The CmPcL cassette was amplified from an existing strain in the laboratory collection, and the synthetic adhesins were amplified from plasmid carrying the already-made constructs kindly sent by I. H. Riedel-Kruse (18). The PCR products were then dialyzed on a 0.025-$\mu$m-porosity filter and electroporated into the *E. coli* K-12 MG1655 pKOBEGA strain. After electroporation, the bacteria were spread on LB plates with the appropriate antibiotic and incubated overnight at 37°C. Once the mutants were verified by PCR, the pKOBEGA plasmid was removed by plating at 42°C and the constructs were transduced using P1vir into a clean genetic background. The strains were then verified a final time by Sanger sequencing. The primers used in this study are listed in Table S9.

**Aggregation curves.** Bacterial cultures were grown overnight in LB medium at 37°C under agitation (180 rpm). One milliliter of overnight culture for each strain was diluted to an optical density at 600 nm (OD$_{600}$) of 3 in spent LB to prevent growth during the experiment under physiologically relevant conditions. At the beginning of the experiment, all the tubes were vortexed and 50 $\mu$L was removed (1 cm below the top of the culture) and mixed with 50 $\mu$L of LB before measurement of the OD$_{600}$. The cultures were left on the bench for 6 h, and the OD$_{600}$ measurement 1 cm below the top of the culture was collected every hour.

**Amikacin survival assay.** Bacteria were grown as described above to an OD$_{600}$ of 0.5 and then left for aggregation at 37°C for 3 h. Bacteria were then treated with 0, 30$\times$, 40$\times$, 50$\times$, or 60$\times$ the MIC of amikacin that was previously determined (2 $\mu$g/mL). The antibiotic solution was added directly into the tube on top of the bacteria and carefully mixed with the culture in order not to break the aggregates. The tubes were then put at 37°C for 18 h. After treatment, the bacteria were washed twice and the aggregates were thoroughly vortexed in order to disrupt a maximum of aggregates. Serial dilutions from $10^{-1}$ to $10^{-6}$ were performed, spotted on LB agar, and placed overnight (14 to 16 h) at 37°C before counting. The results were plotted and statistical analyses were performed using R software (version 4.0.2) implemented in RStudio (version 1.3.1093) using the *ggplot2* (79) and *ggpubr* (80) packages.

**Aggregation experiment and RNA extraction.** Culture sampling, RNA extraction, and RNA-seq were performed in two different batches, each corresponding to the Ag43 and nanobody experiments. Bacteria were grown at 37°C under agitation (180 rpm) in LB medium to an OD$_{600}$ of 0.5 and then transferred directly to separating funnels and left at 37°C for 3 h under static conditions (Fig. S3). Under these conditions, the expected aggregation of cells was strong, while gravity sedimentation of nonaggregating cells was very limited (Fig. S3). Then, by opening the tap of the separating funnel, we collected 1 mL of the lower part of the culture, corresponding to the aggregated cells for strains forming aggregates or to the sedimented cells for nonaggregating strains. Amounts of 150 $\mu$L of the aggregated or nonaggregated cells were washed with 2 volumes of RNAprotect (Qiagen) to prevent RNA degradation and then centrifuged, and the pellets were kept at $-80$°C until extraction. Total RNA was extracted using the MP Biomedicals FastRNA pro blue kit following the provider's manual and treated with the Ambion Turbo DNA-free kit to remove DNA contamination (catalog number AM1907; Thermo Fischer Scientific). Total RNAs from 4 independent replicates for each strain/couple were quantified using the Qubit RNA high-sensitivity (HS) assay (Invitrogen), and quality and integrity confirmed on RNA 6000 RNA chips on the Bioanalyzer system (Agilent).

**Microscopy.** During sampling of bacteria prior to RNA extraction, a drop of each type of aggregate was spotted on a microscopy glass slide (Superfrost plus; Thermo Scientific) and covered with a coverslip. The slides were then left on the bench for 15 min in order to let the aggregates settle. Twenty pictures of each type of aggregate were then taken using an epifluorescence microscope with the phase-contrast mode (EVOS M7000; Invitrogen). The images were then analyzed using FIJI software (version 2.9.0). Using a macro repeating the exact same action for each image, they were smoothed to improve selection of

aggregates over isolated cells, the background was set to black and converted to a mask, and then, using the analyze particle and the measure functions, the number of aggregates and their areas were determined and imported into Excel.

**RNA sequencing.** RNA samples from the Ag43 experiment were subjected to the QIAseq FastSelect −5S/16S/23S kit (Qiagen) for rRNA depletion according to the manufacturer's instructions. The NEBNext bacteria rRNA depletion kit (New England Biosciences) was used for the nanobody experiment RNA samples. No significant difference in rRNA removal efficiency was observed between the 2 sample sets. The libraries from both experiments were generated using the TruSeq stranded total RNA library preparation kit (Illumina, USA) following the manufacturer's protocol. Library quality control was performed on an Agilent Bioanalyzer. Multiplexed RNA sequencing was performed on the Illumina NextSeq 500 platform to generate 150-bp paired-end reads. Sequencing was performed to a depth of 8 million reads per sample for each of 16 samples per experiment (32 total).

**Bioinformatic analysis.** The RNA-seq analysis was performed with Sequana (81). In particular, we used the RNA-seq pipeline (version 0.11.0) (https://github.com/sequana/sequana_rnaseq) built on top of Snakemake 6.1.1 (82). Briefly, reads were trimmed from adapters using Fastp 0.20.1 with default settings (83) and then mapped to the *Escherichia coli* strain K-12 substrain MG1655 genome assembly ASM584v2, accession number GCF_000005845.2 from NCBI, using Bowtie2 (version 2.4.2) (84). FeatureCounts 2.0.1 (85) was used to produce the transcript count matrix, assigning reads to features using corresponding annotation from NCBI guided by strand-specific information. Quality control statistics were summarized using MultiQC 1.10.1 (86). Statistical analysis of the count matrix was performed to identify differentially regulated genes by performing 8 different comparisons of each aggregating sample to its nonaggregating counterpart. Differential expression testing was conducted using DESeq2 1.24.0 (87), indicating the significance (Benjamini-Hochberg-adjusted $P$ value and false discovery rate [FDR] of <0.05) and the effect size (fold change) for each comparison.

The raw transcript counts per gene were imported into RStudio and normalized using the variance-stabilizing transformation (VST) in the DESeq2 library 1.24.0 (87). The principal-component analysis (PCA) was calculated from VST-normalized counts using prcomp() from base stats version 4.0.2 and then visualized using autoplot() from the package ggfortify version 0.4.13 (88). In order to compare genes commonly regulated between aggregate and nonaggregate comparisons, we extracted a gene list per comparison containing all genes where the $\log_2$ fold change ($|\log_2 FC|$) was >1 and the adjusted $P$ value ($P$adj) was <0.05. These gene lists were joined and visualized using the upsetR package (version 1.4.0) (89). The significantly regulated genes common to all comparisons or unique to each experiment were extracted as gene lists and imported into the PANTHER bioinformatics web server (http://www.pantherdb.org) (Tables S1, S2, S4, and S6) (24, 90). The GO groupings that were over- or underrepresented in each experimental group (common to all or unique to each Ag43 or nanobody experiment) were calculated based on the numbers of significantly regulated genes relative to what would be expected by chance. All processed sequencing files and R scripts used to produce the figures in the manuscript are available on Zenodo (91).

**Data availability.** The raw sequencing data for this study have been deposited in the European Nucleotide Archive (ENA) at EMBL-EBI under accession number E-MTAB-11396. All other raw data, processed sequencing files, and scripts to reproduce the figures in the manuscript are available in the Zenodo repository: (https://doi.org/10.5281/zenodo.7595094) (91).

## SUPPLEMENTAL MATERIAL

Supplemental material is available online only.

**SUPPLEMENTAL FILE 1**, XLSX file, 0.9 MB.
**SUPPLEMENTAL FILE 2**, XLSX file, 0.4 MB.
**SUPPLEMENTAL FILE 3**, XLSX file, 0.01 MB.
**SUPPLEMENTAL FILE 4**, XLSX file, 1.2 MB.
**SUPPLEMENTAL FILE 5**, XLSX file, 0.01 MB.
**SUPPLEMENTAL FILE 6**, XLSX file, 0.3 MB.
**SUPPLEMENTAL FILE 7**, XLSX file, 0.01 MB.
**SUPPLEMENTAL FILE 8**, PDF file, 0.7 MB.

## ACKNOWLEDGMENTS

This work was supported by grants from the French Government's Investissement d'Avenir program, Laboratoire d'Excellence Integrative Biology of Emerging Infectious Diseases (grant no. ANR-10-LABX-62-IBEID) and by the Fondation pour la Recherche Médicale (grant no. DEQ20180339185). Y.C. was supported by a MENESR (Ministère Français de l'Education Nationale, de l'Enseignement Supérieur et de la Recherche) fellowship. R.J.S. was supported by a grant from the Philippe Foundation. We would like to thank Joana De Sousa for assistance creating Figure 7.

Biomics Platform -C2RT- was supported by France Génomique (grant no. ANR-10-INBS-09-09) and IBISA.

We declare no conflict of interest.

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
