## [Reviewer comments · Microbiology Spectrum]

Microbiology Spectrum

Escherichia coli aggregates mediated by native or synthetic adhesins exhibit both core and adhesin-specific transcriptional responses

Christophe Beloin, YANKEL CHEKLI, Rebecca Stevick, Etienne KORNOBIS, Valérie Briolat, and Jean-Marc GHIGO

Corresponding Author(s): Christophe Beloin, Institut Pasteur

Review Timeline:

Submission Date:	February 17, 2023
Editorial Decision:	March 7, 2023
Revision Received:	March 10, 2023
Accepted:	March 20, 2023

Editor: Olga Soutourina

Reviewer(s): The reviewers have opted to remain anonymous.

Transaction Report:

DOI: <https://doi.org/10.1128/spectrum.00690-23>

March 7, 2023

Dr. Christophe Beloin
Institut Pasteur
Paris
France

Re: Spectrum00690-23 (Escherichia coli aggregates mediated by native or synthetic adhesins exhibit both core and adhesin-specific transcriptional responses)

Dear Dr. Christophe Beloin:

Thank you for submitting your manuscript to Microbiology Spectrum.

We have now received the comments from two reviewers (provided together with this letter), both of them found the work interesting and elegantly designed. Please consider the concerns raised by the reviewers for the revision step, in particular to explain the medium used and the additional control under shaking conditions.

Link Not Available

Sincerely,

Olga Soutourina

Journals Department
Reviewer comments:

Reviewer #1 (Comments for the Author):

The article by Chekli et al. studies how aggregation alters gene expression in E. coli. To do this, they design two alternative aggregation systems: one based on the constitutive expression of the main E. coli adhesin (ag43), and another based on the expression in some bacteria of an antigen and in other cells of a nanobody that recognises this antigen. The aggregates formed by these bacteria are compared at the transcriptomic level with bacteria that do not express Ag43 or that express a nanobody and an antigen that is not recognised by the nanobody.

The results show much more expression changes when aggregation is induced by the native protein than when bacteria aggregate expressing the nanobody.

The article raises an interesting question and takes an imaginative approach to answering it.

My biggest criticism of the study comes from the way the bacteria are grown and harvested for RNA purification and transcriptome analysis. The authors grow the bacteria in static conditions and purify the aggregates that settle to the bottom of the tube. Non-aggregating bacteria also settle to the bottom, but to a lesser extent, and given that the bacteria are grown to an OD of 0.5, the difference in time spent at the bottom of the tube by aggregating bacteria compared to non-aggregating bacteria may influence the results obtained. I understand that Ag43-mediated and also nanobody induced aggregation also occurs when bacteria are incubated under agitated conditions. If this is the case, it would be interesting to compare the transcriptome of the same bacterial strains grown under shaking conditions. This would help to know whether the results obtained are influenced by the anoxic conditions (microaerophilia) that cause sedimentation and are inherent to the aggregation process.

Reviewer #2 (Comments for the Author):

Chekli et al. present a manuscript examining the effects of autoaggregation of bacteria on the physiology of the bacteria within aggregates. They do this elegantly by transcriptomics, cleverly using a separation funnel to isolate aggregated bacteria. The results show how the bacteria become more resistant to a variety of stresses, a phenomenon that has been known for bacterial aggregates for some time. The authors further examine the effects of native (Ag43) and synthetic (nanobody-based) autoagglutinins and find that the native autoagglutinin shows an enhanced response. As the authors rightly point out, how general this enhanced response is remains to be seen. Other open questions include finding out how the autoaggregation signal is transduced from the cell surface to elicit the transcriptional responses. Another question is where in the autoaggregates these changes take place - throughout or mainly towards the centre of the aggregate. These questions are clearly beyond the scope of the current manuscript, which is well written and nicely presented. I only have some minor comments to address:

1. The authors could provide an explanation for why amikacin was specifically chosen for the survival assays.
2. In figure 1, the authors should follow the nomenclature suggested by Drobnak et al. (2015) regarding autotransporter regions. So 'passenger' rather than alpha domain, etc. The authors should also note that Intimin includes a D00 domain between the beta-barrel and the D0 domain (see Weikum et al., 2020).
3. The authors used spent medium to prevent growth during their autoaggregation experiments. However, spent medium may contain compounds that are toxic or have an effect on cellular physiology (e.g. Quorum sensing molecules). The authors do of course have controls, but even so, they could justify the use of spent medium over say PBS, which would also prevent growth.
4. For the microscopy to determine aggregate size, it is not clear to me how the cells were made fluorescent. Please add this information.

Staff Comments:

Preparing Revision Guidelines

Please return the manuscript within 60 days; if you cannot complete the modification within this time period, please contact me. If you do not wish to modify the manuscript and prefer to submit it to another journal, please notify me of your decision immediately so that the manuscript may be formally withdrawn from consideration by Microbiology Spectrum.

Response to Reviewers Spectrum00690-23

Reviewer #1 (Comments for the Author):

The article by Chekli et al. studies how aggregation alters gene expression in *E. coli*. To do this, they design two alternative aggregation systems: one based on the constitutive expression of the main *E. coli* adhesin (ag43), and another based on the expression in some bacteria of an antigen and in other cells of a nanobody that recognises this antigen. The aggregates formed by these bacteria are compared at the transcriptomic level with bacteria that do not express Ag43 or that express a nanobody and an antigen that is not recognised by the nanobody.

The results show much more expression changes when aggregation is induced by the native protein than when bacteria aggregate expressing the nanobody.

The article raises an interesting question and takes an imaginative approach to answering it.

Response:

Thank you to the reviewer for the positive feedback.

My biggest criticism of the study comes from the way the bacteria are grown and harvested for RNA purification and transcriptome analysis. The authors grow the bacteria in static conditions and purify the aggregates that settle to the bottom of the tube. Non-aggregating bacteria also settle to the bottom, but to a lesser extent, and given that the bacteria are grown to an OD of 0.5, the difference in time spent at the bottom of the tube by aggregating bacteria compared to non-aggregating bacteria may influence the results obtained. I understand that Ag43-mediated and also nanobody induced aggregation also occurs when bacteria are incubated under agitated conditions. If this is the case, it would be interesting to compare the transcriptome of the same bacterial strains grown under shaking conditions. This would help to know whether the results obtained are influenced by the anoxic conditions (microaerophilia) that cause sedimentation and are inherent to the aggregation process.

Response:

The bacteria were not grown in static conditions but in shaking conditions (180 rpm). We apologize because this was not clearly indicated, and we have now added this information in the M&M section (see below)

Line 587: "Bacterial cultures were grown overnight in LB medium at 37°C under agitation (180 rpm).

This is true that aggregation can occur when cells are under agitation, but this aggregation remains limited, and in this condition (without settlement) it would have been impossible to properly collect the aggregating cells. In our hands, clearly, the aggregation is maximal when bacteria are left to settle for some time under static conditions. This is why we chose to limit our analysis on bacteria grown under agitation then left to settle for 3h. In this condition, the deposition of non-aggregative cells due to gravity is very limited, if not inexistent, as shown in the funnels presented in Figure S3A compared to aggregative cells in Figure S3B.

Since the sampling of both aggregating and non-aggregating cells have been performed in a funnel, if some sedimentation of non-aggregating cells occurred, then the sedimented cells have also been collected. Consequently, our analysis reflects the transcriptional response of actively aggregating cells as compared to (very limited) gravity sedimented cells.

We have also added more details on these different aspects in the M&M section:

Line 608-614: "Bacteria were grown at 37°C under agitation (180 rpm) in LB medium until $OD_{600} = 0.5$, then were transferred to separating funnels and left at 37°C for 3 hours in static conditions (Supplementary Fig. S3). In these conditions, the expected aggregation of cells

was strong while gravity sedimentation of non-aggregating cells was very limited (supplementary Fig. S3). Then, by opening the tap of the separating funnel, we collected 1mL of the lower part of the culture, corresponding to the aggregated cells for strains forming aggregates or to the sedimented cells for non-aggregating strains. »

Reviewer #2 (Comments for the Author):

Chekli et al. present a manuscript examining the effects of autoaggregation of bacteria on the physiology of the bacteria within aggregates. They do this elegantly by transcriptomics, cleverly using a separation funnel to isolate aggregated bacteria. The results show how the bacteria become more resistant to a variety of stresses, a phenomenon that has been known for bacterial aggregates for some time. The authors further examine the effects of native (Ag43) and synthetic (nanobody-based) autoagglutinins and find that the native autoagglutinin shows an enhanced response. As the authors rightly point out, how general this enhanced response is remains to be seen. Other open questions include finding out how the autoaggregation signal is transduced from the cell surface to elicit the transcriptional responses. Another question is where in the autoaggregates these changes take place - throughout or mainly towards the centre of the aggregate. These questions are clearly beyond the scope of the current manuscript, which is well written and nicely presented. I only have some minor comments to address:

Response:

Thank you to the reviewer for the positive feedback and for taking up the questions that our work raises.

1. The authors could provide an explanation for why amikacin was specifically chosen for the survival assays.

Response:

To validate that the aggregates formed in our study display the characteristic enhanced survival to external stress, we decided to use a bactericidal antibiotic. Aminoglycosides including amikacin are recommended for the treatment of Enterobacterales infections (see for example: Albert O, *et al.* Antibiotic lock therapy for the conservative treatment of long-term intravenous catheter-related infections in adults and children: When and how to proceed? *Guidelines for clinical practice 2020. Infectious diseases now* **51**, 236-246 (2021) ; Ipekci T, *et al.* Clinical and bacteriological efficacy of amikacin in the treatment of lower urinary tract infection caused by extended-spectrum beta-lactamase-producing *Escherichia coli* or *Klebsiella pneumoniae*. *Journal of infection and chemotherapy : official journal of the Japan Society of Chemotherapy* **20**, 762-767 (2014) ; Pitta RD, *et al.* Antimicrobial therapy with aminoglycoside or meropenem in the intensive care unit for hospital associated infections and risk factors for acute kidney injury. *European journal of clinical microbiology & infectious diseases : official publication of the European Society of Clinical Microbiology* **39**, 723-728 (2020)).

We have added this rationale to the text on lines 138-139:

Aggregated cells survived the lethal action of amikacin, an antibiotic recommended for the treatment of Enterobacterales infections(20-22).

2. In figure 1, the authors should follow the nomenclature suggested by Drobnak et al. (2015) regarding autotransporter regions. So 'passenger' rather than alpha domain, etc. The authors

should also note that Intimin includes a D00 domain between the beta-barrel and the D0 domain (see Weikum et al., 2020).

Response:

We have modified the nomenclature accordingly.

3. The authors used spent medium to prevent growth during their autoaggregation experiments. However, spent medium may contain compounds that are toxic or have an effect on cellular physiology (e.g. Quorum sensing molecules). The authors do of course have controls, but even so, they could justify the use of spent medium over say PBS, which would also prevent growth.

Response:

We used spent medium to dilute the cultures when evaluating the capacity of aggregation of cells carrying the different constructions (Fig. 1C aggregation curves). Despite the possible presence of some molecules that can impact on bacterial physiology we consider that this is still more physiological than resuspending the cells in PBS. This has been justified in the text on line 594: 1 mL of overnight cultures for each strain were diluted to $OD_{600} = 3$ in spent LB to prevent growth during the experiment **under physiologically relevant conditions**.

However, dilution in spent medium was not performed for the main experiments of this manuscript where we collected the aggregated vs non-aggregated cells and performed the transcriptomic analysis. In this later case aggregate formation occurred directly in cultures at $OD = 0.5$ to avoid any interference or influence of non-desired molecules.

4. For the microscopy to determine aggregate size, it is not clear to me how the cells were made fluorescent. Please add this information.

Response:

The aggregates were observed using an epifluorescence microscope but using the phase contrast mode. The cells used were not made fluorescent.

We have now added this information in the M& M section:

“line 637-638: “20 pictures of each type of aggregates were then taken using epifluorescence microscope **with the phase contrast mode** (EVOS M7000, Invitrogen). Images were then analyzed using FIJI software (Version 2.9.0) ».

March 20, 2023

Dr. Christophe Beloin
Institut Pasteur
Paris
France

Re: Spectrum00690-23R1 (Escherichia coli aggregates mediated by native or synthetic adhesins exhibit both core and adhesin-specific transcriptional responses)

Dear Dr. Christophe Beloin:

Thank you for submitting a revised version of the manuscript to Microbiology Spectrum journal.

I am pleased to inform you that your manuscript has been accepted, and I am forwarding it to the ASM Journals Department for publication. You will be notified when your proofs are ready to be viewed.

Sincerely,

Olga Soutourina
Editor, Microbiology Spectrum
